# Elevated Serum Levels of miRNA-155 in Children with Atopic Dermatitis: A Potential Biomarker of Disease

**DOI:** 10.3390/ijms26199689

**Published:** 2025-10-04

**Authors:** Natalia Gołuchowska, Aldona Ząber, Sylwia Walczewska, Agata Będzichowska, Klaudia Brodaczewska, Aleksandra Majewska, Bolesław Kalicki, Agata Tomaszewska

**Affiliations:** 1Department of Pediatrics, Nephrology and Allergology, Military Institute of Medicine—National Research Institute, Szaserów 128, 04-141 Warsaw, Polandabedzichowska@wim.mil.pl (A.B.); awawrzyniak@wim.mil.pl (A.T.); 2Laboratory of Molecular Oncology and Innovative Therapies, Military Institute of Medicine—National Research Institute, Szaserów 128, 04-141 Warsaw, Poland; 3Faculty of Medicine, University of Warsaw, Żwirki i Wigury 61, 02-091 Warsaw, Poland

**Keywords:** atopic dermatitis, eczema, miRNA-155, miRNA-224, miRNA-100, biomarkers, inflammation

## Abstract

Atopic dermatitis (AD) is the most common inflammatory skin disease in the pediatric population. In recent years, the role of microRNAs in inflammatory and immunological mechanisms as specific biomarkers of AD has received growing attention. The aim of the present study was a quantitative assessment of serum expression levels of miR-100, miR-224 and miR-155 in children with AD compared with healthy peers, and an analysis of their potential associations with clinical disease phenotype, severity of skin lesions (SCORAD), cytokine profile, immunological parameters and the presence of concomitant allergic diseases. The study included 12 children with AD and 9 healthy children. Selected miRNAs were isolated from serum, followed by reverse transcription using universal primers and quantification by qRT-PCR. Children with AD exhibited significantly higher expression levels of miR-155 compared with controls (*p* = 0.003). No statistically significant differences were observed for miR-100 and miR-224. miR-100 expression was significantly higher in children with a positive history of inhalant allergy compared with those without such a diagnosis (*p* = 0.014). A positive correlation was observed between miR-100 levels and the percentage of eosinophils (r = 0.599; *p* = 0.052) as well as absolute eosinophil count (r = 0.600; *p* = 0.051). MiR-155 is significantly upregulated in children with AD suggesting it as a candidate biomarker worthy of further investigation in larger cohorts. Although miR-100 did not differentiate the groups, its correlation with eosinophilia and inhalant allergy suggests a role in disease phenotyping.

## 1. Introduction

Atopic dermatitis (AD) is a chronic, relapsing, inflammatory skin disease. Its main symptoms include persistent, intense itching and clinically appearing eczema skin lesions [1]. It is characterized by epidermal barrier dysfunction, dysregulation of the immune response and frequent comorbidity with other atopic diseases. In recent years there has been an observed increase in the prevalence of AD, particularly in highly developed countries. It is estimated that this disease affects 10–20% of the pediatric population and 2–10% of adults worldwide [2]. AD can develop at any age, however, the first symptoms most commonly appear before the age of 5 [3]. In a significant number of children (40–80%), the disease tends to resolve spontaneously before reaching school age. Nevertheless, in one-fifth of patients, AD persists into adulthood [3]. Diagnosis is based on the Hanifin and Rajka criteria [4]. AD has the highest burden of disease (BOD) among skin diseases, with a DALY (disability-adjusted life years) of 5.62 million in the year 2021 [5,6]. The mechanism of AD is not yet fully understood, however, research highlights several key factors contributing to the development of this disease. One significant factor is the malfunction of the immune system. Particularly important is Th2 cell-mediated inflammation, which plays a central role in the disease process [2]. Upon activation, Th2 lymphocytes secrete a specific set of cytokines, among which the most important are interleukin 4 (IL-4), interleukin 13 (IL-13), and interleukin 5 (IL-5). These inflammatory mediators serve a dual function in the pathogenesis of AD. On one hand, they stimulate the production of immunoglobulin E (IgE) by B lymphocytes, characteristic of allergic reactions. On the other hand, they directly affect keratinocytes, leading to disturbances in the expression of epidermal structural proteins, thereby exacerbating epidermal barrier dysfunction [7,8]. Interleukin 6 (IL-6) and interleukin 10 (IL-10) may also play roles in the pathogenesis of AD. IL-6 acts as both a pro-inflammatory and anti-inflammatory cytokine [9]. Studies have demonstrated that IL-6 participates in the development of autoimmune and chronic inflammatory diseases, such as inflammatory bowel disease, diabetes, rheumatoid arthritis, multiple sclerosis, and asthma [9,10,11,12,13,14]. IL-10 is an anti-inflammatory cytokine primarily produced by T lymphocytes, B lymphocytes, macrophages, monocytes, and keratinocytes. Research suggests that IL-10 also possesses immunostimulatory properties, functioning to eliminate infectious and allergic agents. It has a significant impact on the progression of inflammatory, neoplastic, and autoimmune diseases [15]. In recent years, increasing attention has been given to the role of microRNAs (miRNAs) in inflammatory and immunological mechanisms. miRNAs are short, single-stranded, non-coding RNA molecules consisting of approximately 19–25 nucleotides [16]. By modulating gene expression, they play a crucial role in various biological processes, such as proliferation, differentiation, and apoptosis [17,18]. In clinical practice, the analysis of specific biomarkers in plasma or serum is a fundamental diagnostic, prognostic, and therapeutic monitoring tool. Aberrations in miRNA concentrations have been identified across a range of conditions, including inflammatory and immunological dermatoses, contributing to a deeper understanding of their etiopathogenesis and the development of innovative diagnostic and therapeutic strategies [18,19,20,21]. Particular attention has been directed towards miRNA-155, miRNA-224, and miRNA-100, which perform regulatory functions in inflammatory pathways, Th2-type responses, and immune cell differentiation. Studies suggest that miRNA-155 has a bidirectional effect on the Th2 response, potentially both promoting and inhibiting it through interactions with various transcription factors, and the regulation of cytokines such as IL-4, IL-5, IL-10, and IL-13 [22,23,24,25].

Additionally, miRNA-155 affects the expression of CTLA-4 (cytotoxic T lymphocyte-associated protein 4), a key regulator of T lymphocyte activation which can modulate inflammatory processes in AD [26]. 

miRNA-100 exhibits regulatory effects on cell proliferation, differentiation, and metabolism. In the context of AD, it acts anti-inflammatorily by suppressing the NLR family pyrin domain containing 3 signaling pathway [27,28]. 

Studies in allergic rhinitis reveal that reduced miRNA-224 expression in nasal mucosa is associated with increased levels of cyclin-dependent kinase 9 (CDK9) and pro-inflammatory cytokines. Similarly, in asthma animal models, upregulation of miRNA-224 attenuates inflammation and airway remodeling. The mechanism involves inhibition of TLR2 and suppression of Th17-derived inflammatory mediators, highlighting the anti-inflammatory potential of miRNA-224 in airway diseases [29,30,31].

The aim of this study was to evaluate the expression of miRNA-100, miRNA-224, and miRNA-155 in the serum of children with AD compared to healthy peers, and to analyze their potential associations with the clinical phenotype of the disease, severity of skin lesions (SCORAD), cytokine profile, immunological parameters, and the presence of concomitant allergic diseases.

## 2. Results

### 2.1. Group Characteristics

The study included a total of 21 children. The study group consisted of 12 patients with AD, including 8 girls and 4 boys, with an average age of 5.83 years (±3.51). The control group comprised 9 healthy children without signs of allergic diseases (6 girls and 3 boys) with an average age of 8.56 years (±4.27).

No significant differences were found between the groups regarding basic anthropometric parameters. The mean height in the study group was 115.79 ± 25.02 cm, while in the control group, it was 131.89 ± 19.45 cm (*p* = 0.126). The growth percentile was 50.00 ± 29.88 in the AD group and 55.22 ± 25.02 in the control group (*p* = 0.668). The body weight was higher in the control group (median: 31.30 kg, IQR: 17.40–45.30) than in the AD group (20.03 kg, IQR: 13.49–27.62); however, this difference was not statistically significant (*p* = 0.111). Similarly, the difference in body weight percentiles (AD: 41.83 ± 28.02 vs. control: 62.22 ± 23.30) did not reach statistical significance (*p* = 0.085).

Laboratory studies showed a significantly higher total leukocyte count (WBC) in children with AD compared to the control group (median: 8.71 × 10^9^/L, IQR: 7.72–13.44 vs. 6.42 × 10^9^/L, IQR: 5.27–7.55; *p* = 0.009). Despite having a higher percentage and absolute count of eosinophils in patients with AD (percentage: 6.40%, IQR: 3.40–12.70 vs. 4.10% IQR: 3.25–8.40; absolute count: 0.66 × 10^3^/µL, IQR: 0.26–1.50 vs. 0.27 × 10^3^/µL, IQR: 0.16–0.69), these differences did not reach statistical significance (*p* = 0.345 and *p* = 0.148, respectively).

The total concentration of immunoglobulin E was significantly elevated in the AD patient group (median: 487.50 IU/mL, IQR: 123.75–1151.00) compared to the control group (45.00 IU/mL, IQR: 34.50–66.00; *p* = 0.012) (Table 1).

The clinical profile indicated a highly severe disease course, with a median SCORAD score of 60 points (IQR 56.00–72.00). All patients received emollient therapy as the basic treatment, and 12 patients required additional topical anti-inflammatory treatment with corticosteroids in combination with calcineurin inhibitors.

Atopic diseases often present with multiple comorbidities. In the studied group, nine children had diagnosed respiratory allergies and seven had food allergies. In 25% (n = 3) of the participants concomitant asthma was observed, while 67% (n = 8) exhibited allergic rhinitis (Table 1).

This profile reflects the typical phenotype of pediatric patients with severe AD-characterized by multiple atopic comorbidities and the frequent need for long-term topical therapy.

### 2.2. Evaluation of miRNA in the Pathogenesis of AD

Among the analyzed miRNAs, a statistically significant higher expression was noted for miRNA-155 in children with AD compared to the control group (0.109 RE, IQR: 0.079–0.230 vs. 0.052 RE, IQR: 0.015–0.109; *p* = 0.003).

No statistically significant differences were observed in the levels of miRNA-100 (0.039 RE, IQR: 0.023–0.073 vs. 0.029 RE, IQR: 0.010–0.067; *p* = 0.272) or miRNA-224 (0.143 RE, IQR: 0.086–0.311 vs. 0.162 RE, IQR: 0.075–0.297; *p* = 1.000), although a tendency toward higher expression of miRNA-100 in AD patients was noted (Table 2, Figure 1).

Analysis of selected pro- and anti-inflammatory cytokines did not reveal significant differences between the study groups. The IL-4 concentration was lower in the AD group (0.00 ng/mL, IQR 0.00–26.05) compared to the control group (13.60 ng/mL, IQR 0.00–38.50), but this difference was not statistically significant (*p* = 0.473). Similarly, IL-13, a key cytokine of the Th2 axis, exhibited a wide range of values with no group differences (median 118.50 ng/mL, IQR 23.56–825.11 vs. 475.55 ng/mL, IQR 65.75–618.55; *p* = 0.536).

Levels of IL-5, IL-6, and IL-10 remained low or undetectable in both groups, with no significant differences observed (respectively, *p* = 0.536; *p* = 0.295; *p* = 0.657).

To assess the influence of concomitant atopic diseases on miRNA expression in the group of children with AD, independent statistical analyses were conducted concerning the presence of food and respiratory allergies (Figure 2).

No significant differences were observed in the expression levels of miRNA-100, miRNA-224, or miRNA-155 between children with and without food allergies (*p* = 0.201; *p* = 0.372; *p* = 0.570, respectively).

In contrast, analysis of the impact of respiratory allergy revealed a significantly higher expression of miRNA-100 in children with a positive history of respiratory allergy compared to those without such a diagnosis (0.050 RE, IQR 0.030–0.097 vs. 0.023 RE, IQR 0.018–0.020; *p* = 0.014). However, the expression levels of miRNA-224 (*p* = 0.309) and miRNA-155 (*p* = 0.644) did not differ significantly depending on the presence of respiratory allergy.

### 2.3. Assessment of Factors Influencing miRNA in AD

In the analysis of the relationships between the expression of examined miRNAs and selected clinical and immunological markers, it was demonstrated that miRNA-100 may play a particular role in the immunopathogenesis of AD. The expression of miRNA-100 showed a negative correlation with the SCORAD index (r = –0.612; *p* = 0.060), suggesting that lower levels of this molecule may be associated with greater severity of skin lesions. Simultaneously, a positive association was observed between miRNA-100 levels and eosinophil percentage (r = 0.599; *p* = 0.052) as well as their absolute count (r = 0.600; *p* = 0.051), which may indicate the involvement of this miRNA in eosinophilic inflammatory response mechanisms.

For miRNA-155, no statistically significant correlations were observed with any of the analyzed clinical parameters, including SCORAD, eosinophilia, total IgE, or Th2 cytokines. Similarly, miRNA-224 expression remained independent of cytokine levels, eosinophil count, and disease severity.

Figure 3 presents the strength of the correlation (r) between miRNA expression and clinical parameters.

Analysis of the impact of topical corticosteroids (tCS) and topical calcineurin inhibitors (tCI) on the expression of miRNA-100, miRNA-224, and miRNA-155 revealed no statistically significant differences between children using these topical treatments and those not undergoing such therapy (Table 3).

## 3. Discussion

AD is the most common chronic inflammatory skin disease. Its pathogenesis is not yet fully understood, and among the key factors believed to contribute to the development of this condition are abnormalities in the immune system function [7].

In this study, the expression levels of three selected miRNAs—miRNA-155, miRNA-224, and miRNA-100—were analyzed in the serum of children with AD in comparison to a group of healthy peers. The results indicate a significantly higher expression level of miRNA-155 in the patient group, with no statistically significant differences observed in the levels of miRNA-224 and miRNA-100 between the groups.

The influence of miRNA-155 on Th2 cells is complex and inconclusive. Conflicting evidence exists regarding its role as either an inhibitor or an activator of the Th2 response [18,22,23,24,25]. Literature suggests that suppression of miRNA-155 expression supports the Th2 response and enhances the production of IL-4, and IL-10 [32]. Furthermore, miRNA-155 may inhibit Th2 differentiation by interacting with the transcription factor c-Maf [33]. Other studies indicate that higher expression of miRNA-155 can stimulate the secretion of Th2 cytokines, such as IL-5 and IL-13, and that decreased miRNA-155 levels in mice are associated with lower cytokine levels, potentially due to the suppression of the transcription factor PU.1 which inhibits the Th2-dependent response [34,35,36]. The literature indicates that the concentration of miRNA-155 is elevated in individuals with atopic diseases, including serum samples from patients with allergic rhinitis, as well as in respiratory tissues and plasma of asthmatic patients [37,38,39]. In the context of AD, miRNA-155’s ability to modulate the Th2 response is of particular importance, owing to its influence on the expression of transcription factors, such as GATA-3, and its regulation of cytokine production including IL-4, IL-5, and IL-13.

In this study, significantly higher serum levels of miRNA-155 were observed in children with AD compared to healthy controls. Similar findings were reported by Ma et al., who documented that miRNA-155 may contribute to AD pathogenesis through the modulation of Th17 lymphocyte differentiation and function, thereby supporting the persistence of chronic skin inflammation [40]. Additionally, El-Korashi et al. demonstrated that miRNA-155 levels correlate with disease severity in pediatric AD patients, suggesting its potential as a prognostic biomarker for disease activity [41]. Elevated miRNA-155 levels have also been detected in skin biopsies and serum samples of patients with AD, as shown by Sonkoly and colleagues [26].

Furthermore, in in vitro studies using peripheral blood mononuclear cells (PBMCs) and skin biopsies, it was demonstrated that CTLA-4, a negative regulator of T cell activation, is a direct target of miRNA-155. Elevated miRNA-155 resulted in decreased CTLA-4 expression which was accompanied by an increased proliferative response of T lymphocytes [26]. Recent research by Jing Chang et al. further confirmed that serum levels of miRNA-155 are elevated in AD patients [42]. Moreover, in mouse models, they showed that IL-32 promotes AD development by upregulating JAK1 expression, which in turn leads to increased miRNA-155 expression [42].

In our study, miRNA-155 expression did not show significant correlations with serum Th2 cytokine levels or total IgE, which may be attributed to several reasons. First, the lack of correlation could result from the small sample size, limiting statistical power. Second, cytokine levels in serum are highly variable and characterized by a short half-life, making it difficult to capture concurrent changes alongside miRNA levels.

Analysis of the relationships between miRNA profiles and clinical features in this study revealed that miRNA-100 maintains a positive correlation with eosinophil percentage and absolute eosinophil count.

Patients with AD often present with peripheral blood eosinophilia. Eosinophils migrate into the skin in response to chemokines, particularly eotaxins, and accumulate in inflamed areas [43]. Once activated, they release granule contents, which damage keratinocytes and impair the skin barrier. This tissue injury and mediator release amplifies chronic inflammation and facilitates allergen and microbe penetration [44]. In addition, eosinophils secrete cytokines, such as IL-4, IL-13, and TGF-β, which reinforce the Th2 response and promote skin remodeling and fibrosis [45]. It is also known that interaction of eosinophils with nerve endings may contribute to pruritus [46].

It also shows a negative correlation with the SCORAD index, though not reaching statistical significance.

miRNA-100 functions as a regulator of cellular proliferation, differentiation, and metabolism. It plays a critical role in the development and progression of tumors exerting either tumor-suppressive or tumor-promoting effects depending on the cancer type. Although data on miRNA-100 in AD are limited, such a profile of activity aligns with the disease phenotype, which involves barrier deficits, microinflammation, and increased epidermal renewal [27]. Wu et al. demonstrated that miRNA-100 has potential in treating AD. miRNA-100 exerts anti-inflammatory effects by suppressing the expression of forkhead box O3, which in turn inhibits the activation of the downstream NLR family pyrin domain containing 3 signaling pathway [28].

In this work, a significantly higher expression of miRNA-100 was observed in children with a positive history of respiratory allergies compared to those without such a diagnosis. The association of miRNA-100 with respiratory allergy, along with its positive correlation with eosinophilia, suggests its involvement in the regulatory network of type 2 immune response, particularly during eosinophil recruitment and activation. This finding has potentially important implications for understanding the pathogenesis of respiratory allergies in the pediatric population. Clinically, miRNA-100 emerges as a candidate marker for phenotyping respiratory allergy in children with AD. It may also aid in stratifying patients with eosinophilic profiles. From a biomarker perspective, further validation in larger cohorts is recommended.

In summary, in our data, miRNA-100 does not distinctly differentiate between groups but provides a consistent biological signal: it is positively associated with eosinophilia and respiratory allergy. Simultaneously, it shows a non-significant association opposite to clinical severity. This pattern supports the hypothesis that miRNA-100 plays a modulatory role in the skin barrier and in the type 2 inflammatory axis.

miRNA-224 levels have not been evaluated in AD in previous studies yet. In the studied pediatric population with AD, miRNA-224 did not exhibit significant differences in expression compared to the control group. The statistical analysis also revealed no correlations with hematological parameters, Th2 cytokine levels, IgE concentrations, or clinical indices such as SCORAD. The lack of dependence on the presence of respiratory or food allergies and on the use of topical corticosteroids suggests that miRNA-224 does not directly reflect the severity of type 2 inflammation or disease activity within our study model.

In our investigation, no elevated levels of IL-4 and IL-13 were observed between the AD children and the control group, despite these cytokines being commonly described as characteristic of AD. This result indicates a different dynamic of cytokine concentrations in the pediatric population, which may be related to methodological differences, characteristics of the studied groups, or the unique immunopathological processes occurring in children with AD [47]. Levels of IL-5, IL-6, and IL-10 remained low or undetectable in both groups and showed no significant differences.

The absence of expected differences in interleukin concentrations could be influenced by numerous factors. According to studies by Altara et al., cytokine levels can exhibit significant diurnal fluctuations, which may impact results depending on the timing of serum collection [48,49]. Additionally, environmental factors such as diet and stress are known to modulate cytokine concentrations in blood, which could contribute to the lack of observed differences in our analysis [50]. These factors may have contributed to the absence of anticipated cytokine level differences in our study.

Research on miRNA-224 has also been conducted in the context of the pathogenesis of sensitive skin. Sensitive skin is characterized by heightened reactivity to various external stimuli with a defective stratum corneum permeability barrier being a hallmark feature [51]. Previous studies have demonstrated that increased levels of miRNA-224 in keratinocytes lead to a deficiency of CLDN5, resulting in increased skin permeability in sensitive skin [51].

It is also known that in patients with allergic rhinitis, miRNA-224 levels are reduced in nasal mucosal cells [29]. Further investigations into this relationship were carried out by Sang Wang and colleagues, who developed an artificial neural network model in mice. They observed decreased miRNA-224 expression in the nasal mucosa of animal models, accompanied by elevated levels of cyclin-dependent kinase 9 (CDK9), which promotes the production of IL-6 and TNF-α. Their study found that increasing miRNA-224 levels in nasal epithelial cells significantly alleviated symptoms of allergic rhinitis, such as nose rubbing and sneezing. Moreover, it decreased serum levels of IL-4, IL-6, and IL-18 in mice with AR, suggesting that the anti-inflammatory effects of miRNA-224 may be linked to the inhibition of Th2 cytokines [30].

Additionally, in mouse model studies, higher levels of miRNA-224 were shown to attenuate asthma symptoms by suppressing inflammation and airway remodeling. It was demonstrated that miRNA-224 in asthma inhibits effects induced by airborne PM2.5 particles by targeting toll-like receptor 2 (TLR2) and suppresses inflammatory mediators secreted by Th17 cells, including IL-17, IL-4, and IL-5, which are part of the Th2 cytokine pathway [31].

In summary, the analyzed group reveals a complementary but heterogeneous profile of the two examined miRNAs. miRNA-155 appears as an indicator of “disease presence” and activation of the inflammatory axis. Conversely, miRNA-100 may serve as a marker of the “allergic-eosinophilic phenotype” and potentially of “lower disease severity” (a negative non-significant association with SCORAD). This delineation supports the concept of a miRNA panel with distinct applications: diagnostic (miRNA-155) and phenotypic (miRNA-100). Such a division of roles justifies the development of multidimensional predictive models integrating these molecules with classical atopic markers.

There are several reasons for the lack of statistical significance in the levels of miRNA-100 and miRNA-244.

The study’s small sample size might not provide enough statistical power to detect subtle changes in miRNA expression. AD is a heterogeneous disease, and the patient population may have variations that influence the results. Future research should explore whether miRNA-224 and miRNA-100 may be more relevant in specific AD subtypes (e.g., early-onset vs. late-onset, intrinsic vs. extrinsic). It is possible the chosen cohort did not capture the relevant patient population where these miRNAs would show significant changes. Furthermore, the analysis focused on serum miRNA levels, potentially missing localized changes in the skin, which is the primary site of AD pathology. Examining miRNA expression in skin biopsies might yield different results for miRNA-224 and miRNA-100.

Despite the significant findings of this study, several limitations must be acknowledged, which could influence the interpretation of the data. First, a key factor restricting the statistical power and generalizability is the relatively small sample size. Second, the analysis was based on a single measurement of markers in serum, which, given the high dynamic variability of cytokines and miRNAs in circulation, may not reflect long-term trends or fluctuations associated with disease exacerbations. Repeated sampling and standardized collection times would be beneficial in future studies. Third, the expression of miRNAs was assessed solely in serum, which may not fully capture local skin processes—the primary site of inflammation.

The fourth limitation of the study was disease severity spectrum, where all patients had severe AD, preventing conclusions across the full clinical range. The fifth limitation is control group imbalance, where the age-matching would improve reliability.

The exploratory nature of this pilot study, combined with the limited sample size and single serum measurements, restricts the ability to generalize the results. The absence of miRNA expression analysis in skin biopsies further limits the conclusions. Therefore, the obtained data should be considered preliminary and require validation in larger, age-matched cohort and inclusion of mild, and moderate AD.

## 4. Material and Methods

### 4.1. Study Population

A total of 21 children were enrolled in the study, including 12 patients with a diagnosis of AD of moderate to severe severity (study group) and 9 healthy children (without allergic diseases, chronic diseases, active infectious disease, congenital or acquired immunodeficiency) as a control group. The diagnosis of AD was based on the current clinical criteria established by Hanifin and Rajka. The severity of AD was assessed using the standardized SCORAD index: mild disease was defined as SCORAD < 25, moderate as 25–50, and severe as ≥50.

Exclusion criteria included the use of systemic glucocorticoids, immunosuppressive agents, or allergen-specific immunotherapy within the last six months, the presence of chronic diseases other than asthma or allergic rhinitis, autoimmune diseases, and dermatoses of other etiologies. The detailed inclusion and exclusion criteria are presented in Table 4.

All participants underwent a detailed medical history assessment and a comprehensive physical examination, which included evaluation of the clinical presentation of AD, coexisting atopic conditions (respiratory allergy, food allergy, asthma, allergic rhinitis), and current topical treatments.

In the subsequent stage, blood samples were collected for the measurement of cytokines (IL-4, IL-13, IL-5, IL-6, and IL-10) and miRNAs (miRNA-100, miRNA-155, and miRNA-224).

The study was approved by the Bioethics Committee of the Military Medical Chamber (1 September 2023, decision no. 23/23). Legal guardians of all participants provided written informed consent for the children’s participation in the study.

### 4.2. Blood Serum Collection

For each blood collection, peripheral blood was drawn into a serum tube (Vacutainer SST™ II Advance, BD, Warsaw, Poland) and allowed to clot for 30 min at room temperature (RT). Blood samples were fractionated by centrifugation at 2000× *g* for 15 min at RT to obtain serum. The supernatants were then aliquoted, frozen and stored at −80 °C until analysis.

### 4.3. miRNA Detection in the Serum

RNA, including small RNAs, from serum was isolated using Plasma/Serum Circulating and Exosomal RNA Purification Kits (cat# 42800 Norgen Biotek Corp, Thorold, ON, Canada) according to the manufacturer’s protocol; 250 µL of thawed serum was used for isolation. Small RNA concentration was checked using the fluorometer Qubit (cat# Q32880, Qubit™ microRNA Assay Kits, Thermo Fisher Scientific, Waltham, MA, USA) according to the manufacturer’s instructions. Equal volumes of RNA isolates, from each sample, were used to obtain cDNA using TaqMan™ Advanced miRNA cDNA Synthesis Kit (cat# A28007 Thermo Fisher Scientific, CA, USA, USA). Briefly, PolyA tailing, adapter ligation and reverse transcription reaction with universal primers were performed. The product of reverse transcription was pre-amplified (with 18 cycles), then diluted 1:10 and used for the qRT-PCR reaction with TaqMan™ Fast Advanced Master Mix and TaqMan™ Advanced miRNA Assay (hsa-miR-155-5p, 483064_mir; has-miR-224-5p 483106_mir; has-miR-100-5p, 478224_mir; Thermo Fisher Scientific, Waltham, MA, USA). Spike-in controls (synthetic cel-miR-55-3p oligonucleotide sequence) was added to serum sample) was used to monitor the effectiveness of miRNA isolation and cDNA synthesis and checked by TaqMan™ Advanced miRNA Assay (cel-miR-55-3p, Thermo Fisher Scientific, Waltham, MA, USA). Reactions were run on Bio-Rad CFX384 qPCR System (BioRad, Hercules, CA, USA) according to the protocol described in Table 5. The relative miRNAs levels were calculated with 2(-Delta C(T)) method, with normalization to the geometric mean of reference miRNAs Ct values: hsa-miR-191-5p (477952_mir, Thermo Fisher Scientific, Waltham, MA, USA) and hsa-miR-484 (478308_mir, Thermo Fisher Scientific, Waltham, MA, USA).

### 4.4. Cytokines Detection in the Serum

Cytokines (IL-4, IL-13, IL-5, IL-6 and IL-10) levels in serum samples were measured using enzyme-linked immunosorbent assay (DuoSet ELISA, R&D System, Minneapolis, MN, USA). The assay was performed according to the manufacturer’s protocol, adapted to 384 well plates. Briefly, serum samples were thawed on ice, centrifuged 2000× *g*, 10 min, 4 °C and 25 µL of sample was tested in duplicates on primary antibody coated and blocked 386 well plates. Detection was performed using HRP substrate, TMB. Optical density at 450 nm, was measured using plate spectrophotometric reader (VarioScan Lux, ThermoFisher Scientific, Waltham, MA, USA). Standard curve for corresponding recombinant protein and blanked absorbance measurements were performed. Results are presented as concentration in reference to standard curve.

### 4.5. Statistical Methods

No formal sample size calculation was performed due to the exploratory nature of this study. The final sample size was determined by the available recruitment capacity and laboratory throughput during the study period.

The statistical analysis was performed using SPSS Statistics for Windows, version 29.0.0.0 (IBM Corp., New York, NY, USA). In all analyses, *p*-values < 0.05 were considered statistically significant. In the first stage of the analysis, the normality of the distribution of variables was assessed using the Kolmogorov–Smirnov test with Lilliefors correction. For variables with a normal distribution, the mean was used as a central measure and the standard deviation as a measure of dispersion. For variables with a non-normal distribution, the median was used as a central measure and the interquartile range as a measure of dispersion. For comparisons of continuous variables in two groups, Student’s *t*-test was used for variables with a normal distribution, and the Mann–Whitney U test was used for variables with a non-normal distribution. For comparisons of dichotomous variables, the chi-square test was used, and in the case of its assumptions not being met, Fisher’s exact test was used. In order to assess the dependence of two continuous variables, correlation calculations were performed.

## 5. Conclusions

Our study demonstrated that miRNA-155 is significantly elevated in children with atopic dermatitis, supporting its potential role as a diagnostic biomarker. Although miRNA-100 did not differentiate between groups, it correlated with eosinophilia and respiratory allergy, suggesting its relevance in phenotyping the disease.

## Figures and Tables

**Figure 1 ijms-26-09689-f001:**
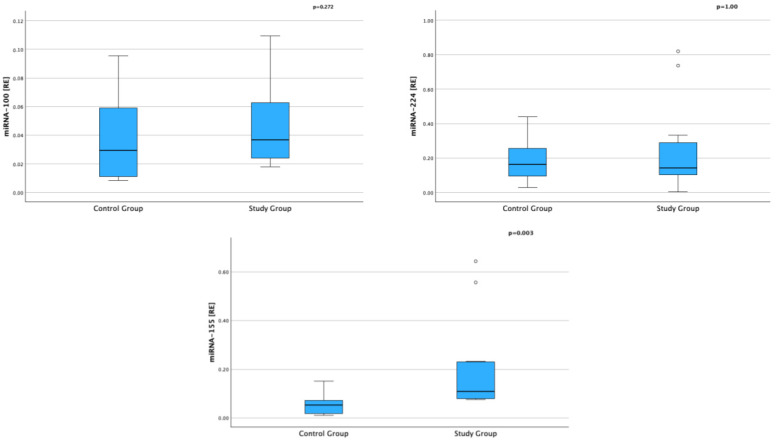
Expression of Selected miRNAs in Serum in Children with AD and in the Control Group (RE—relative expression).

**Figure 2 ijms-26-09689-f002:**
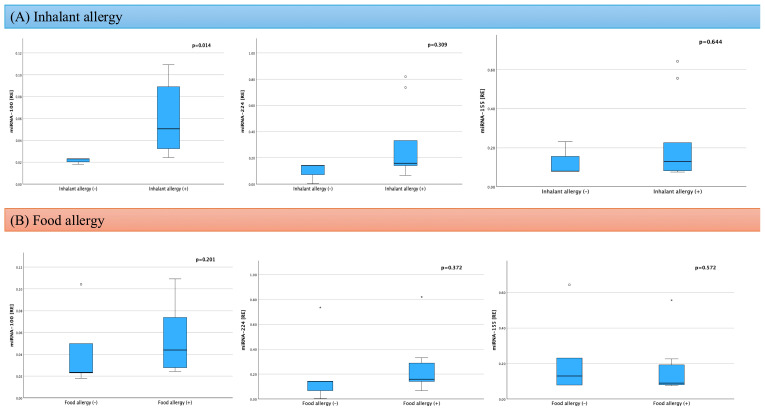
Expression of miRNA-100, miRNA-224, and miRNA-155 in children with AD according to the presence of respiratory allergy (**A**) and food allergy (**B**) [* RE—relative expression].

**Figure 3 ijms-26-09689-f003:**
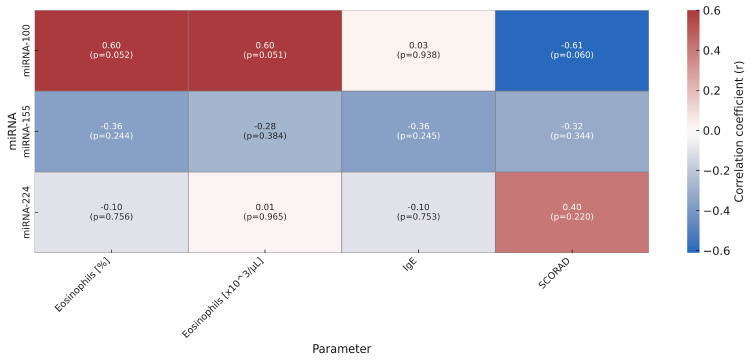
Heatmap of Pearson correlation coefficients between miRNA expression and selected clinical and immunological parameters in children with atopic dermatitis. Each cell displays the correlation coefficient (r) and the corresponding *p*-value. The color scheme indicates the direction and strength of the correlation: from blue (negative) to red (positive).

**Table 1 ijms-26-09689-t001:** Characteristics of the Study and Control Groups.

Variable	Study Group (n = 12)	Control Group (n = 9)	*p*-Value
Age [years]; mean (±SD)	5.83 (±3.51)	8.56 (±4.27)	*p*^St^ = 0.125
Sex; n (%)			
Females	8 (66.7)	8 (66.7)	*p*^chi2^ = 1.00
Males	4 (33.3)	4 (33.3)	
Height [cm]; mean (±SD)	115.79 (25.02)	131.89 (19.45)	*p*^St^ = 0.126
Height [pc]; mean (±SD)	50.00 (29.88)	55.22 (25.02)	*p*^St^ = 0.668
Weight [kg]; median (IQR)	20.03 (13.49–27.62)	31.30 (17.40–45.30)	*p*^UMW^ = 0.111
Weight [pc]; mean (±SD)	41.83 (28.02)	62.22 (23.30)	*p*^St^ = 0.085
WBC [×10^9^/L]; median (IQR)	8.71 (7.72–13.44)	6.42 (5.27–7.55)	*p*^UMW^ = 0.009
Eosinophils [%]; median (IQR)	6.40 (3.40–2.70)	4.10 (3.25–8.40)	*p*^UMW^ = 0.345
Eosinophils [×10^3^/μL]; median (IQR)	0.66 (0.26–1.50)	0.27 (0.16–0.69)	*p*^UMW^ = 0.148
IgE [IU/mL]; median (IQR)	487.50 (123.75–1151.00)	45.00 (34.50–66.00)	*p*^UMW^ = 0.012
Vitamin D [ng/mL]; mean (±SD)	32.96 (10.15)	26.22 (9.05)	*p*^St^ = 0.154
SCCORAD [points]; median (IQR)	60.00 (56.00–72.00)	-	-
Comorbidities [n], %		-	-
Respiratory allergies	9 (75)	-	-
Food allergies	7 (58)	-	-
Asthma	3 (25)	-	-
Allergic Rhinitis	8 (67)	-	-

SD—standard deviation, IQR—interquartile range; St—Student *t*-test, UMW—Mann–Whitney U test, chi2—chi-square test (asymptotic, two-sided), WBC—white blood count, n—number.

**Table 2 ijms-26-09689-t002:** Expression of Selected miRNAs and Cytokine Concentrations in Serum in Children with AD and in the Control Group.

Variable	Study Group (n = 12)	Control Group (n = 9)	*p*-Value
miRNA-100 [RE]; median (IQR)	0.039 (0.023–0.073)	0.029 (0.010–0.067)	*p*^UMW^ = 0.272
miRNA-224 [RE]; median (IQR)	0.143 (0.086–0.311)	0.162 (0.075–0.0297)	*p*^UMW^ = 1.000
miRNA-155 [RE]; median (IQR)	0.109 (0.079–0.230)	0.052 (0.015–0.109)	*p*^UMW^ = 0.003
IL-4 [ng/mL]; median (IQR)	0.00 (0.00–26.05)	13.60 (0.00–38.50)	*p*^UMW^ = 0.473
IL-13 [ng/mL]; median (IQR)	118.50 (23.56–825.11)	475.55 (65.75–618.55)	*p*^UMW^ = 0.536
IL-5 [ng/mL]; median (IQR)	0.00 (0.00–00.62)	0.00 (0.00–0.00)	*p*^UMW^ = 0.536
IL-6 [ng/mL]; median (IQR)	0.00 (0.00–28.30)	12.44 (0.00–310.1555)	*p*^UMW^ = 0.295
IL-10 [ng/mL]; median (IQR)	0.00 (0.00–58.74)	0.85 (0.00–25.80)	*p*^UMW =^ 0.657

RE—relative expression; IQR—interquartile range; UMW—Mann–Whitney U test, n—number.

**Table 3 ijms-26-09689-t003:** Values of Selected microRNAs in the Group of Children with AD Using Topical Glucocorticoids (tCS) and Topical Calcineurin Inhibitors (tCI).

Variable	tCS/tCI (+) (n = 7)	tCS/tCI (−) (n = 5)	*p*-Value
miRNA-100 [RE]; median (IQR)	0.026 (0.022–0.056)	0.051 (0.030–0.106)	*p*^UMW^ = 0.201
miRNA-224 [RE]; median (IQR)	0.143 (0.141–0.737)	0.067 (0.035–0.246)	*p*^UMW^ = 0.167
miRNA-155 [RE]; median (IQR)	0.089 (0.078–0.178)	0.159 (0.079–0.555)	*p*^UMW^ = 0.465

RE—relative to hsa-miR-191-5p and hsa-miR-484 expression, IQR—interquartile range; UMW—Mann–Whitney U test, n—number.

**Table 4 ijms-26-09689-t004:** Inclusion and Exclusion Criteria of the Study Group.

Inclusion Criteria	Exclusion Criteria
Diagnosis of atopic dermatitis based on Hanifin and Rajka criteria	Other skin diseases
Moderate to severe form of AD (SCORAD ≥ 25 points)	Active infectious disease
Consent to participate in the study	Autoimmune diseases
	Congenital or acquired immunodeficiency
	Lack of consent to participate in the study

**Table 5 ijms-26-09689-t005:** qRT-PCR conditions—TaqMan™ Fast Advanced Master Mix (miRNA detection).

Step	Temperature	Time	Cycle
Enzyme activation	95 °C	20 s	1
Denature	95 °C	1 s	40
Anneal/Extend	60 °C	20 s	

## Data Availability

The datasets used during the current case report are available from the corresponding author on reasonable request.

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
