# Peer review of "Elevated Serum Levels of miRNA-155 in Children with Atopic Dermatitis: A Potential Biomarker of Disease"

_ijms, 2025, doi:10.3390/ijms26199689_

Round 1

Reviewer 1 Report

Comments and Suggestions for Authors

The manuscript addresses an important topic: the potential role of circulating microRNAs as biomarkers in pediatric AD. The study is well written, the methodology is clearly described, and the discussion integrates previous findings. 
However, several issues may limit the strength of the conclusions.

  1. Sample size is very small (12 AD vs 9 HC), restricting statistical power. 
  2. Control group imbalance, where the age-matching would improve reliability.
  3. Disease severity spectrum, where all patients had severe AD, preventing conclusions across the full clinical range.
  4. Statistical interpretation, with non-significant associations, is presented as "trends"; these should be removed, in my opinion. Also, no correction for multiple testing was applied...
  5. The absence of IL-4/IL-13 differences contradicts established AD pathophysiology. Standardization of sampling times or repeated measurements should be considered in the cytokine data. Why not consider a flow cytometry-based approach for the cytokine assessment? This would also allow identification of the specific immune cells responsible for cytokine production. 
  6. Why were only three microRNAs analyzed, given that AD is strongly influenced by multiple immunomodulatory microRNAs?
  7. Regarding "Inhalant/Respiratory allergies", do you specifically mean Allergic Asthma and Allergic Rhinitis? If so, it would be valuable to present separate analyses of microRNAs expression for these conditions. This could potentially reveal distinct expression patterns that help differentiate disease profiles within the atopic triad.  

I recommend reframing the paper as a preliminary, exploratory pilot study, with more cautious language regarding biomarker conclusions, especially since the findings were not validated using complementary techniques. In addition, a larger, age-matched cohort and inclusion of mild and moderate AD would be necessary to validate these findings.

Minor issues:

  1. Figure 2 would be clearer with a larger font size and improved image resolution.
  2. A few typos (e.g., "gourdians") should be corrected. 
  3. I also suggest numbering the lines in the manuscript to facilitate precise feedback.

Author Response

  1. Reviewer’s comment: Sample size is very small (12 AD vs 9 HC), restricting statistical power. 

Author’s response: Thank You for pointing this out. We agree that the small sample size in our study limited the statistical power. As this was a pilot investigation, we intend to expand our research with a larger cohort in future studies to address this limitation and strengthen our findings. You can find annotation about this in the limitations of the study (discussion, page 11-12, lines 355-371).

  1. Reviewer’s comment: Control group imbalance, where the age-matching would improve reliability.

Author’s response: Thank You for this comment. We acknowledge the reviewer's point regarding the control group imbalance and the importance of age-matching for improved reliability. In future studies, we will prioritize more precise age-matching in our control group selection. You can find annotation about this in the limitations of the study (discussion, page 11-12, lines 355-371).

  1. Reviewer’s comment: Disease severity spectrum, where all patients had severe AD, preventing conclusions across the full clinical range.

Author’s response: Thank you for this comment. We acknowledge the reviewer's valid point regarding the disease severity spectrum. The fact that our pilot study focused exclusively on patients with severe AD limits the generalizability of our conclusions across the full clinical range of the disease. Future research will aim to include participants representing a broader spectrum of AD severity, allowing for a more comprehensive understanding of the observed effects. You can find annotation about this in the limitations of the study (discussion, page 11-12, lines 355-371).

  1. Reviewer’s comment: Statistical interpretation, with non-significant associations, is presented as "trends"; these should be removed, in my opinion. Also, no correction for multiple testing was applied...

Author’s response: We thank the Reviewer for this important observation. We agree that interpretation of non-significant results as “trends” may be misleading. In the revised version we have removed such wording and now describe these findings as non-significant associations. Regarding multiple testing correction, we acknowledge that numerous comparisons were performed, which increases the risk of type I error. Given the exploratory nature of our study and the small sample size, we decided not to apply strict corrections, as this could mask potentially relevant biological signals.

  1. Reviewer’s comment: The absence of IL-4/IL-13 differences contradicts established AD pathophysiology. Standardization of sampling times or repeated measurements should be considered in the cytokine data. Why not consider a flow cytometry-based approach for the cytokine assessment? This would also allow identification of the specific immune cells responsible for cytokine production. 

Author’s response: We thank the Reviewer for this insightful comment. We agree that IL-4 and IL-13 play a central role in AD pathophysiology and that the absence of differences in our serum data may appear unexpected. As noted by the Reviewer, serum cytokine concentrations are highly variable and influenced by multiple external and temporal factors. In our study, only single time-point measurements were performed, which may have limited the sensitivity for detecting subtle group differences. We have added this point to the Limitations section and emphasized that repeated sampling and standardized collection times would be beneficial in future studies. You can find annotation about this in the limitations of the study (discussion, page 11-12, lines 355-371).

Unfortunately, for this project we could obtain only serum samples and only ELISA tests could be purchased. We are aware of the limitations of this approach and in future project, involving bigger group of patients we will include more sensitive tests and cytometry analysis of immune cells. Thank you for the suggestion of repeated measurements, indeed this could help with the cytokine measurements reliability.

  1. Reviewer’s comment: Why were only three microRNAs analyzed, given that AD is strongly influenced by multiple immunomodulatory microRNAs?

Author’s response: In this preliminary study, only three selected miRNAs were examined, as it was a pilot investigation. We plan to conduct further research involving a larger panel of miRNAs in patients with atopic dermatitis, with the aim of gaining a more comprehensive understanding of their roles and potential as biomarkers or therapeutic targets.

  1. Reviewer’s comment: Regarding "Inhalant/Respiratory allergies", do you specifically mean Allergic Asthma and Allergic Rhinitis? If so, it would be valuable to present separate analyses of microRNAs expression for these conditions. This could potentially reveal distinct expression patterns that help differentiate disease profiles within the atopic triad.  

Author’s response: In the inhalant allergy section, we simply meant inhalant allergy. In future studies, we intend to present separate analyses of microRNA expression for asthma and allergic rhinitis. The above work focused only on atypic dermatitis.

  1. Reviewer’s comment: I recommend reframing the paper as a preliminary, exploratory pilot study, with more cautious language regarding biomarker conclusions, especially since the findings were not validated using complementary techniques. In addition, a larger, age-matched cohort and inclusion of mild and moderate AD would be necessary to validate these findings.

Author’s response: Thank you for raising this point. I agree with the reviewer's assessment regarding the classification of our study as a preliminary, exploratory pilot investigation. We acknowledge the limitations inherent in this type of study and, as the reviewer suggested, have framed our conclusions in a more cautious manner in the discussion section. We also emphasize the need for future validation, including using complementary techniques, a larger, age-matched cohort, and the inclusion of patients with mild and moderate atopic dermatitis to further confirm these findings. You can find annotation about this in the limitations of the study (discussion, page 11-12, lines 355-371).

Minor issues:

  1. Reviewer’s comment: Figure 2 would be clearer with a larger font size and improved image resolution.

Author’s response: Thank You for this comment. We have improved the figure 2. according to your comments.

  1. Reviewer’s comment: A few typos (e.g., "gourdians") should be corrected. 

Author’s response: Thank You for this comment. I checked and corrected typos.

  1. Reviewer’s comment: I also suggest numbering the lines in the manuscript to facilitate precise feedback.

Author’s response: Thank You for this comment. I numbered the lines of the manuscript.

Reviewer 2 Report

Comments and Suggestions for Authors

In the introduction, clinical picture of atopic dermatitis is not eczema-like but literally it is eczema. Change the sentence into .... clinically appearing eczema skin lesions. 

The function and role of miRNA in pathogenesis of AD is extremely sparsely explained. Please elaborate in detail and provide extensive explanation with mechanisms and possibly one illustration. 

Can You elaborate small sample size, did You perform statistic prior to research to define appropriate sample size that would be eligible with population. 

Why You did not separate AD groups into extrinsic and intrinsic, or were all of the participants with extrinsic AD, and why. What are possible implications of that to Your results? As I noticed You only compared and divided according to concomitant allergic disease but i just do not understand how You compared theese groups and what sense does this commparison make. 

In disscussion:

”First, miRNA-155 might predominantly reflect immunological activity within the skin rather than in the systemic circulation.”- please elaborate this, as we all know that AD in systemic immunological reaction just with its repercussion in the skin, i do not this this is explanation for the lack of signiciant difference and small sample size is rather the culprit. 

"Analysis of the relationships between miRNA profiles and clinical features in this study revealed that miRNA-100 maintains a positive correlation with eosinophil percentage and absolute eosinophil count"- can You please elaborate this, this is where the problem of not including intrinsic AD arises. As we all know that eosinophils do not have major role in pathogenesis of AD, they do contribute in terms of secreting IL-12 and IL-15 for example, and shifting disease into chronic Th17 and Th22 state with lichenification, but are not the main culprit. In other hand they are major for allergic rhinitis and asthma. This is why I think the results of this study are not valuable enough if You do not show and separate the intrinsic AD group. 

Can ypu also elaborate this: "Conversely, miRNA-100 may serve as a marker of the “allergic-eosinophilic phenotype” and potentially of “lower disease severity” (a negative trend with SCORAD)" This is an interesting finding that I also encountered in my research. Please provide pathomechanism theory and explanation. 

"no elevated levels of IL-4 and IL-13 were observed between the AD children and the control group, despite these cytokines being commonly described as characteristic of AD"- please explain this in more detail. Provide more explanation, because this is an interesting and surprising result. 

Author Response

  1. Reviewer’s comment: In the introduction, clinical picture of atopic dermatitis is not eczema-like but literally it is eczema. Change the sentence into .... clinically appearing eczema skin lesions. 

Author’s response: Thank You for this comment. You can find changed sentence in the introduction section (introduction, page 1, lines 37-38).

  1. Reviewer’s comment: The function and role of miRNA in pathogenesis of AD is extremely sparsely explained. Please elaborate in detail and provide extensive explanation with mechanisms and possibly one illustration. 

Author’s response: Thank you for this valuable comment. To address Yours concern about the limited explanation of the function and role of miRNAs in the pathogenesis of AD, we have added detailed information and expanded on the discussion of the specific miRNAs studied, focusing on their mechanisms of action and relevance to AD. We believe these additions will significantly strengthen the manuscript. You can find it in introduction and discussion section (introduction, pages 2-3, lines 76-97, discussion, pages 8-11).

  1. Reviewer’s comment: Can You elaborate small sample size, did You perform statistic prior to research to define appropriate sample size that would be eligible with population. 

Author’s response: Thank you for this important point. We acknowledge that the small sample size is a limitation of our study. We recognize this and, therefore, have framed the study as a preliminary pilot investigation. We appreciate the suggestion to perform a statistical power analysis prior to future research to determine the appropriate sample size needed to achieve statistical significance, and this will be a crucial aspect of the design for our planned larger, follow-up study.

  1. Reviewer’s comment: Why You did not separate AD groups into extrinsic and intrinsic, or were all of the participants with extrinsic AD, and why. What are possible implications of that to Your results? As I noticed You only compared and divided according to concomitant allergic disease but i just do not understand how You compared theese groups and what sense does this commparison make. 

Author’s response: Thank you very much for this comment. We did not perform a division into extrinsic and intrinsic AD groups due to the small sample size of our study. However, we plan to conduct further research on larger groups, where we will aim to differentiate children with atopic and non-atopic AD. Additionally, it is worth noting that in children, atopic dermatitis is primarily considered to be extrinsic in nature.

  1. Reviewer’s comment: ”First, miRNA-155 might predominantly reflect immunological activity within the skin rather than in the systemic circulation.”- please elaborate this, as we all know that AD in systemic immunological reaction just with its repercussion in the skin, i do not this this is explanation for the lack of signiciant difference and small sample size is rather the culprit. 

Author’s response: Thank you for this valuable comment. The reviewer is indeed correct that the main limitation of our study was the small sample size, which may have impacted the ability to detect significant differences. In future studies, we plan to increase the sample size to enhance the statistical power and obtain more conclusive results. We have reworded this paragraph in the article

  1. Reviewer’s comment: "Analysis of the relationships between miRNA profiles and clinical features in this study revealed that miRNA-100 maintains a positive correlation with eosinophil percentage and absolute eosinophil count"- can You please elaborate this, this is where the problem of not including intrinsic AD arises. As we all know that eosinophils do not have major role in pathogenesis of AD, they do contribute in terms of secreting IL-12 and IL-15 for example, and shifting disease into chronic Th17 and Th22 state with lichenification, but are not the main culprit. In other hand they are major for allergic rhinitis and asthma. This is why I think the results of this study are not valuable enough if You do not show and separate the intrinsic AD group. 

Author’s response: We thank the Reviewer for this valuable comment. We agree that eosinophils are not the main drivers of AD pathogenesis, although they represent an element of the type 2 immune axis in some patients. Recent studies emphasize the variability and limited universal value of eosinophilia as a biomarker in AD, as well as the differences between blood and skin tissue activity. Therefore, in the revised Discussion we clarified that the positive associations between miRNA-100 and eosinophilia should be interpreted as reflecting an atopic/respiratory allergy–related phenotype rather than a mechanistic driver of AD itself.

We also acknowledge that the pilot character of our study and the limited sample size prevented us from reliably distinguishing between extrinsic and intrinsic AD phenotypes. This point has now been explicitly added to the Limitation section. 

In future studies, we plan to standardize sampling times, apply longitudinal designs, and consider cell-based approaches (e.g., flow cytometry) or tissue-level analyses, as suggested by the Reviewer.

  1. Reviewer’s comment: Can you also elaborate this: "Conversely, miRNA-100 may serve as a marker of the “allergic-eosinophilic phenotype” and potentially of “lower disease severity” (a negative trend with SCORAD)" This is an interesting finding that I also encountered in my research. Please provide pathomechanism theory and explanation. 

Author’s response: Thank You for this comment. You can find annotation about this in the discussion section (discussion, page 10, lines 267-275).

  1. Reviewer’s comment: "no elevated levels of IL-4 and IL-13 were observed between the AD children and the control group, despite these cytokines being commonly described as characteristic of AD"- please explain this in more detail. Provide more explanation, because this is an interesting and surprising result. 

Author’s response: We thank the Reviewer for this insightful comment. As we know, the IL-4 and IL-13 play a central role in AD pathophysiology and that the absence of differences in our serum data may appear unexpected. Serum cytokine concentrations are highly variable and influenced by multiple external and temporal factors. In our study, only single time-point measurements were performed, which may have limited the sensitivity for detecting subtle group differences. We have added this point to the Limitations section and emphasized that repeated sampling and standardized collection times would be beneficial in future studies.

You can find annotation about this in the limitations of the study (discussion, page 11-12, lines 355-371).

Reviewer 3 Report

Comments and Suggestions for Authors

The manuscript "Elevated Serum Levels of miRNA-155 in Children with Atopic Dermatitis: A Potential Biomarker of Disease" presents an interesting investigation into the search for new biomarkers for atopic dermatitis (AD) in children. The study of serum microRNAs is a relevant and timely field. The main finding of elevated miR-155 levels in AD patients is potentially important and aligns with the known role of this miRNA in immune processes. However, in its current form, the manuscript has several serious methodological and substantive shortcomings that must be addressed prior to acceptance for publication.

The major points requiring revision are:

  1. The introduction provides a general overview of AD but lacks depth in justifying the selection of these three specific miRNAs (miR-155, miR-224, miR-100). Why were these chosen over other candidates? What are the known functions of these miRNAs in the context of skin immunity, T-helper cell differentiation (Th1/Th2), or barrier function? It is critical to add a paragraph with a brief review of existing data (including references) on the role of each investigated miRNA in allergic and inflammatory diseases, particularly AD, to substantiate the study's hypothesis.

  2. The sample size is extremely small (n=12 in the AD group, n=9 in the control). This is the primary limitation of the study, significantly reducing its statistical power and the reliability of its conclusions. The inclusion/exclusion criteria for the control group are not specified (e.g., the absence of any allergic or chronic diseases?).

    • Suggestion:

      • Explicitly state in the "Limitations" section that the study is pilot in nature and the small sample size is its main constraint.

      • Describe the selection criteria for the control group in detail.

  3. While the use of SCORAD is mentioned, data on disease severity within the cohort (mean ± standard deviation, range) are not provided. This is an essential characteristic of the patient population. Data on SCORAD must be presented in the "Results" section or in a patient characteristics table.

  4. The method description lacks sufficient detail for reproducibility. The commercial RNA extraction kit is not named, and there is no mention of RNA quality/quantity control (e.g., using an Agilent Bioanalyzer). The specific universal primers and commercial kit for cDNA synthesis are not indicated. For qRT-PCR, the detection system (SYBR Green / TaqMan) and amplification protocol (annealing temperatures, cycle number) are missing. The most critical omission is the information regarding which reference miRNA (or housekeeping genes) was used for data normalization. Without this, the expression data are invalid.

  5. The results are presented clearly, but their interpretation is hindered by methodological gaps. The absence of a table with patient characteristics (age, sex, SCORAD, IgE level, comorbidities) makes it impossible to assess the representativeness and comparability of the groups.

  6. The discussion is logically focused on miR-155, but it could be strengthened. The negative results for miR-224 and miR-100 are insufficiently discussed. Why were they unchanged? Could their role be more specific to other AD phenotypes or disease stages?

    • Suggestion:

      • Expand the discussion by linking the miR-155 findings to its well-known role in Th2 lymphocyte polarization and the production of IL-4, IL-5, and IL-13, which is key to AD pathogenesis.

      • Discuss potential reasons for the negative results concerning the other miRNAs.

  7. The conclusions are overly strong for a study with such a small sample size. The wording needs to be softened. For example, instead of "supporting its potential diagnostic utility," a more appropriate phrasing would be "suggesting its potential as a candidate biomarker worthy of further investigation in larger cohorts."

In my opinion, the article requires major revision. After addressing the noted shortcomings—especially those in the "Materials and Methods" section, adding a patient characteristics table, and tempering the conclusions to reflect the study's limitations—the manuscript could be resubmitted for further review. The research topic is of interest to the journal's readership, and the preliminary data are promising.

Author Response

  1. Reviewer’s comment: The introduction provides a general overview of AD but lacks depth in justifying the selection of these three specific miRNAs (miR-155, miR-224, miR-100). Why were these chosen over other candidates? What are the known functions of these miRNAs in the context of skin immunity, T-helper cell differentiation (Th1/Th2), or barrier function? It is critical to add a paragraph with a brief review of existing data (including references) on the role of each investigated miRNA in allergic and inflammatory diseases, particularly AD, to substantiate the study's hypothesis.

Author’s response: A detailed description of selected miRNAs can be found in the discussion section (discussion, pages 8-11, lines 217-354). MiRNA-155, miRNA-224, and miRNA-100 were selected based on a literature review, because they perform regulatory functions in inflammatory pathways, Th2-type responses, and immune cell differentiation. In this preliminary study, only three selected miRNAs were examined, as it was a pilot investigation. We plan to conduct further research involving a larger panel of miRNAs in patients with atopic dermatitis, with the aim of gaining a more comprehensive understanding of their roles and potential as biomarkers or therapeutic targets.

  1. Reviewer’s comment: The sample size is extremely small (n=12 in the AD group, n=9 in the control). This is the primary limitation of the study, significantly reducing its statistical power and the reliability of its conclusions. The inclusion/exclusion criteria for the control group are not specified (e.g., the absence of any allergic or chronic diseases?).
    • Suggestion:
      • Explicitly state in the "Limitations" section that the study is pilot in nature and the small sample size is its main constraint.
      • Describe the selection criteria for the control group in detail.

Author’s response: Thank you for raising this point. I agree with the classification of our study as a preliminary, exploratory pilot study. We acknowledge the limitations inherent in this type of study, such as small sample size as its main constraint. You can find annotation about this in the discussion section (discussion, page 11-12, lines 355-371). To describe the selection criteria for the control group, criteria in detail were added to material and methods section (Material and methods, page 12, lines 374-376).

  1. Reviewer’s comment: While the use of SCORAD is mentioned, data on disease severity within the cohort (mean ± standard deviation, range) are not provided. This is an essential characteristic of the patient population. Data on SCORAD must be presented in the "Results" section or in a patient characteristics table.

Author’s response: Thank you for this valuable comment. I have updated Table 1. according to the Reviewer's suggestions.

  1. Reviewer’s comment: The method description lacks sufficient detail for reproducibility. The commercial RNA extraction kit is not named, and there is no mention of RNA quality/quantity control (e.g., using an Agilent Bioanalyzer). The specific universal primers and commercial kit for cDNA synthesis are not indicated. For qRT-PCR, the detection system (SYBR Green / TaqMan) and amplification protocol (annealing temperatures, cycle number) are missing. The most critical omission is the information regarding which reference miRNA (or housekeeping genes) was used for data normalization. Without this, the expression data are invalid.
  1. Reviewer’s comment: The results are presented clearly, but their interpretation is hindered by methodological gaps. The absence of a table with patient characteristics (age, sex, SCORAD, IgE level, comorbidities) makes it impossible to assess the representativeness and comparability of the groups.

Author’s response: Thank you for this valuable comment. We have updated Table 1. according to the Reviewer's suggestions.

  1. Reviewer’s comment: The discussion is logically focused on miR-155, but it could be strengthened. The negative results for miR-224 and miR-100 are insufficiently discussed. Why were they unchanged? Could their role be more specific to other AD phenotypes or disease stages?
    • Suggestion:
      • Expand the discussion by linking the miR-155 findings to its well-known role in Th2 lymphocyte polarization and the production of IL-4, IL-5, and IL-13, which is key to AD pathogenesis.
      • Discuss potential reasons for the negative results concerning the other miRNAs.

Author’s response: Thank you for this insightful comment. I agree that the discussion of miR-155 could be strengthened by linking it to its well-established role in Th2 lymphocyte polarization and the production of IL-4, IL-5, and IL-13, which are central to AD pathogenesis (discussion, pages 8-9, lines 217-244). Additionally, I appreciate the suggestion to provide a more comprehensive discussion of the negative results concerning miR-224 and miR-100, including potential reasons for their lack of change and their possible relevance in other AD phenotypes or disease stages. I addressed all of it in discussion section (discussion, page 11, lines 343-354).

  1. Reviewer’s comment: The conclusions are overly strong for a study with such a small sample size. The wording needs to be softened. For example, instead of "supporting its potential diagnostic utility," a more appropriate phrasing would be "suggesting its potential as a candidate biomarker worthy of further investigation in larger cohorts."

Author’s response: Thank you very much to the reviewer for this valuable comment. You are right that, with such a small sample size, we should be cautious in drawing conclusions. After revising the manuscript, I have made the necessary adjustments accordingly.

Round 2

Reviewer 1 Report

Comments and Suggestions for Authors

The revised version of your manuscript shows clear improvements. I appreciate that you have addressed the main concerns raised in the previous review, especially by:

  • Explicitly acknowledging the limitations (sample size, age imbalance in controls, restriction to severe AD, single-point serum measurements, and lack of skin tissue analysis).
  • Reframing the study as a preliminary, exploratory pilot investigation with appropriately cautious conclusions.
  • Correcting the overinterpretation of non-significant findings and improving figure quality and typos.

The methodological description is now clear, and the discussion better contextualizes the findings within current literature. While the inherent limitations remain, I believe the revised manuscript is substantially strengthened and suitable for publication as a pilot study, provided that the exploratory nature is kept clear to readers.

Author Response

Dear Reviewer,

Thank you very much for your thorough and positive feedback on the revised version of our manuscript. I appreciate your kind words regarding the improvements made, especially in addressing the main concerns such as acknowledging the study's limitations, reframing it as a preliminary investigation, and enhancing the clarity of the methodological description and discussion.

Your constructive comments have been invaluable in strengthening our work.

Thank you again for your valuable insights and support.

Reviewer 2 Report

Comments and Suggestions for Authors

Thank You for the revised version. 

Author Response

Dear Reviewer,

Thank you very much for your thorough and positive feedback on the revised version of our manuscript. Your constructive comments have been invaluable in strengthening our work. Thank you again for your valuable insights and support.

Reviewer 3 Report

Comments and Suggestions for Authors

The authors have revised the manuscript. It can be published in its current form.

Author Response

Dear Reviewer,

Thank you very much for your review of my work. I have addressed your comments by improving the quality of the tables and figures and English. I appreciate your feedback.